# The Effects of Microstructure on the Dynamic Mechanical Response and Adiabatic Shearing Behaviors of a Near-α Ti-6Al-3Nb-2Zr-1Mo Alloy

**DOI:** 10.3390/ma16041406

**Published:** 2023-02-07

**Authors:** Haisheng Chen, Fang Hao, Shixing Huang, Jing Yang, Shaoqiang Li, Kaixuan Wang, Yuxuan Du, Xianghong Liu, Xiaotong Yu

**Affiliations:** 1Western Superconducting Technologies Co., Ltd., Xi’an 710018, China; 2National & Local Joint Engineering Laboratory for Special Titanium Alloy Processing Technologies, Xi’an 710018, China; 3Xi’an Key Laboratory of Special Titanium Alloy Processing and Simulation Technologies, Xi’an 710018, China; 4School of Materials Science and Chemical Engineering, Xi’an Technological University, Xi’an 710018, China; 5State Key Laboratory for Advanced Metals and Materials, University of Science and Technology Beijing, Beijing 100083, China

**Keywords:** titanium alloy, microstructure, dynamic response, adiabatic shearing

## Abstract

The formation and evolution of adiabatic shear behaviors, as well as the corresponding mechanical properties of a near-Ti-6Al-3Nb-2Zr-1Mo (Ti-6321) alloy during dynamic compression process, were systematically investigated by the split Hopkinson pressure bar (SHPB) compression tests in this paper. Ti-6321 samples containing three types of microstructures, i.e., equiaxed microstructure, duplex microstructure and Widmanstätten microstructure, were prepared to investigate the relationship between microstructures and dynamic mechanical behaviors under different strain rates in a range from 1000 s^−1^ to 3000 s^−1^. It was found by the dynamic strain–stress relation that the Ti-6321 alloys containing equiaxed microstructure, duplex microstructure and Widmanstätten microstructure all exhibited a strong strain-hardening effect. The samples containing equiaxed microstructure exhibited a larger flow stress than samples containing duplex microstructure and Widmanstätten microstructure. The adiabatic shearing behaviors in Ti-6321 alloy are significantly influenced by different types of microstructures. The formation of adiabatic shearing bands occurs in equiaxed microstructure when the strain rate is increased to 2000 s^−1^. The adiabatic shear bands are formed in duplex microstructure when the strain rate reaches 3000 s^−1^. However, the initiation of adiabatic shear bands is found in Widmanstätten microstructure under the strain rate of 1000 s^−1^. The Widmanstätten microstructure shows a larger sensitivity to adiabatic shearing than the equiaxed microstructure and duplex microstructure samples.

## 1. Introduction

Near-α titanium alloys have been widely applied to the aerospace, ship engineering and chemical industries due to their low density, high specific strength, excellent corrosion resistance and good weldability [1]. It has been reported that titanium undergoes an allotropic transformation from a close-packed hexagonal structure (α-phase) to a body-centered cubic structure (β-phase) at 882 °C [2,3]. Generally, titanium alloys containing different microstructures can be formed after adding alloy elements to pure titanium, and can be divided into three categories: α, α + β, and metastable β alloys, according to their microstructures [4].

Ti-6Al-3Nb-2Zr-1Mo (Ti-6321) alloy is a representative near-α titanium alloy which has been successfully applied to deep-sea engineering and ship fabrication [5,6,7]. The mechanical properties of titanium alloys are closely related to the morphology of their microstructures. The near-α titanium alloys behave four typical microstructures: equiaxed, duplex, basket weave and Widmanstätten microstructures [8]. The investigation of the relationship between the microstructures and dynamic mechanical properties of Ti-6321 alloy is of great importance and significance, considering that the application of this alloy is related to service with external loads under high strain rates [9].

The split Hopkinson pressure bar (SHPB) test is an effective method to evaluate the dynamic mechanical properties of materials [10,11,12,13]. During the impact process, a uniform and uniaxial loading is imparted on the tested specimen. In recent years, the SHPB technique has been widely used to evaluate the dynamic mechanical properties of titanium alloys [14,15,16]. Previous studies have reported that the flow stress of titanium alloys can be improved significantly due to the the strain-hardening effect. Yu et al. studied the flow behaviors of bimodal TC17 alloy and pointed out that the flow stress was distinctly improved when the strain rate was increased from 1 s^−1^ to 20 s^−1^ due to the variation of strain hardening [17]. Xu et al. studied the dynamic compression properties and deformation mechanism of a Ti-6321 alloy with a duplex structure and found that the strength of the Ti-6321 alloy was increased significantly with increasing strain rates due to the strain-rate hardening effect [18]. It is well known that mechanical properties are closely related to the microstructure of titanium alloys. Therefore, more detailed investigations are required on the influence of microstructure on the dynamic mechanical properties and the corresponding mechanisms of titanium alloys.

Adiabatic shearing is a unique localized failure behavior when metallic materials deform at the high strain rates [19,20]. Generally, deformation under the high strain rates can be considered as a process with the formation of adiabatic shearing areas. The instantaneous severe plastic deformation induces high thermal energy which cannot be conducted immediately. Therefore, the gathered thermal energy could result in a distinct temperature rise in the localized area which also leads to the thermal softening as well as the localized thermal-plastic instability. Titanium and its alloys are adiabatic-shear-sensitive materials due to their small specific heat capacity and low thermal conductivity [21]. Therefore, adiabatic shearing behaviors in titanium alloys have been intensively investigated. It has been pointed out that the α + β lamellar microstructure was more susceptive to adiabatic shearing than β phase microstructure at the same strain rate. Arab and co-authors reported the effects of microstructure on the dynamic properties of TA15 alloy [14]. Yang et al. have reported the effects of microstructure on the adiabatic shearing behaviors of Ti-1300 alloy [22]. They pointed out that the TA15 sample with a bimodal structure exhibited a smaller sensibility to adiabatic shearing than the sample with martensitic α structure. Previous studies have indicated that adiabatic shearing behaviors in titanium alloys can be highly influenced by the microstructure. However, the relationships between microstructure and adiabatic shearing behaviors in titanium alloys also need more investigations and further clarification.

In this study, the dynamic mechanical response of a near-α Ti-6Al-3Nb-2Zr-1Mo (Ti-6321) alloy was systematically investigated using the split Hopkinson pressure bar (SHPB) test. Ti-6321 samples with equiaxed microstructure (EM), duplex microstructure (DM) and Widmanstätten microstructure (WM) were prepared for dynamic mechanical properties testing under different strain rates. The relationships between microstructure and adiabatic shearing behaviors in Ti-6321 alloy were discussed in detail.

## 2. Experimental Material and Procedures

The Ti-6Al-3Nb-2Zr-1Mo (Ti-6321) alloy (bar with a diameter of 85 mm) used in this study was produced by the Western Superconducting Technologies Co., Ltd., Xi’an, China. The real chemical composition of the as-received Ti-6321 alloy was determined and is listed in Table 1.

The as-received Ti-6321 bar was forged into bars with a smaller diameter of 45 mm (at 940 °C, upsetting and swaging three times with a total deformation of about 70%). In order to ensure microstructural uniformity, the forged bar was annealed at 900 °C for 1.5 h and then air-cooled to room temperature. To obtain the target microstructure, appropriate heat treatments were conducted according to the T_β_ of the Ti-6321 alloy. Detailed heat treatments are listed in Table 2.

The dynamic mechanical properties of the Ti-6321 alloy were tested using split Hopkinson pressure equipment. Samples prepared for dynamic mechanical testing were machined to a size of Φ 4 mm × 4 mm. The Hopkinson equipment and size of sample are shown in Figure 1. To explore the dynamic mechanical behaviors of the Ti-6321 alloy under different loading, the tested strain rates were set as 1000 s^−1^, 2000 s^−1^, 3000 s^−1^ and 4000 s^−1^, respectively. Meanwhile, the experimental data were collected through the stress waveform.

The specimens were mechanically sanded with a series of SiC sand paper (600#, 1000#, 2000#, 3000# and 5000#) and vibrationally polished with diamond paste for the microstructure observation. Specimens for microstructural observations were mechanically polished first and then chemically etched using a mixture containing nitric acid, hydrofluoric acid and water with a volume proportion of 1:3:7. Microstructural observations were performed using an OLYMPUS/PMG3 light optical microscope and a scanning electron microscope (SEM) (JSM6700F, Hitachi, Japan).

## 3. Results and Discussion

### 3.1. Microstructural Characterization

Figure 2 exhibits the initial microstructure of the Ti-6321 samples under different heat-treated conditions by the optical observation. The microstructure of the Ti-6321 alloy has been significantly influenced by heat treatments. It can be found that the sample heat-treated at 800 °C shows the equiaxed microstructure (EM) composed of primary α phase and reserved β phase, as shown in Figure 2a. The formation of EM is mainly due to the fact that 800 °C is relatively lower than the primary temperature α phase growth and spheroidization of the Ti-6321 alloy at 1001.5 °C of the β transition temperature according to the phase equilibrium [2,3]. Meanwhile, some β phase is reserved as well. The sample that was heat-treated at 960 °C shows a duplex microstructure (DM) composed of equiaxed primary α grains and transition of β phase, as shown in Figure 2b, since 960 °C is close to the critical transition temperature of T_β_ (1001.5 °C) of the Ti-6321 alloy. The reserved β phase is found transformed into α phase during the air-cooling process. The sample heat-treated at the temperature of 1020 °C above the T_β_ exhibits the typical Widmanstätten microstructure (WM). It was found that several colonies with different orientations formed in the same β grain, as shown in Figure 2c. Moreover, continuous grain-boundary α phase formed along the grain boundary of primary β grains. X-ray diffraction analysis was also performed to confirm the phase component of the Ti-6321 alloy under different heat-treated conditions, as shown in Figure 2d. Both α and β peaks were found in the diffraction patterns of all samples indicating the constitution of the α and β phase.

### 3.2. Dynamic Mechanical Response

The dynamic mechanical properties of Ti-6321 alloy were obtained by compression tests using split Hopkinson pressure equipment under the strain rate of 1000 s^−1^, 2000 s^−1^, 3000 s^−1^ and 4000 s^−1^. Figure 3 shows the macro photographs of the samples with different microstructures after the Hopkinson compression tests. It can be seen that the deformation of all samples is significantly increased with the rise in strain rate. The EM and DM samples exhibited the same case of deformation under each strain rate, as shown in Figure 3a–d. Both the EM and DM samples only showed the height reduction under the strain rates of 1000 s^−1^ and 2000 s^−1^. However, when the strain rate increased to 3000 s^−1^ and 4000 s^−1^, significant height reduction as well as the macro damage of the EM and DM samples occurred under the impact loading. The damage modes were found to be related to the shear fracture with the shear direction of 45° deviation to the axis of the sample. The WM sample showed a height reduction under the strain rate of 1000 s^−1^ and exhibited both a height reduction and 45° shear fracture when the strain rate increased to 2000 s^−1^. Moreover, a larger height reduction and severer shear deformation occurred in the WM sample when the strain rate was increased to 3000 s^−1^ and 4000 s^−1^. Observations on the morphology of the samples indicate that the Ti-6321 alloy with equiaxed or duplex microstructure exhibits stronger resistance to dynamic pressure loading and shearing fracture.

Figure 4a–c show the true stress–strain curves of the Ti-6321 samples with different microstructures under the strain rates of 1000 s^−1^, 2000 s^−1^, 3000 s^−1^ and 4000 s^−1^, respectively. For the EM samples, the dynamic flow stress at the uniform plastic deformation stage was found to be distinctly increased with the increasing strain rates. However, the EM samples exhibited similar flow stress under the strain rates of 3000 s^−1^ and 4000 s^−1^. The dynamic flow stress of DM samples also improved as the strain rate increased, except that the samples showed similar flow stress under the strain rates of 2000 s^−1^ and 3000 s^−1^. The WM samples also exhibited improved dynamic flow stress with the increase in strain rate. Similar to the DM sample, the WM samples showed the similar flow stress under the strain rate of 2000 s^−1^ and 3000 s^−1^ as well. According to the true stress–strain curves, it can be concluded that the Ti-6321 alloy samples containing equiaxed, duplex and Widmanstätten microstructures all exhibit strain-rate sensitivity and the strain-hardening effect.

To evaluate and compare the resistance of the Ti-6321 alloy with EM, DM and WM to the same dynamic loading, the true stress–strain curves of the EM, DM and WM samples were plotted in the same layout according to each strain rate, as shown in Figure 5. It can be seen that the EM, DM and WM samples exhibit different mechanical behaviors under the strain rates of 1000 s^−1^, 2000 s^−1^, 3000 s^−1^ and 4000 s^−1^, respectively. Under the strain rate of 1000 s^−1^, the EM sample shows a larger flow stress than the DM and WM samples during the uniform plastic deformation process. The DM sample exhibits a moderate flow stress whereas the WM sample shows the smallest flow stress. Under the strain rate of 2000 s^−1^, the EM sample also shows the largest flow stress during the uniform plastic deformation process. The DM sample shows a smaller flow stress than the EM sample, whereas the WM sample shows the smallest flow stress. Under the strain rate of 3000 s^−1^, the EM, DM and WM samples exhibit a similar trend in the evolution of flow stress compared with those under the strain rate of 2000 s^−1^. The EM sample shows the largest flow stress during the uniform plastic deformation process. The DM sample exhibits a moderate flow stress whereas the WM sample shows the smallest flow stress. Under the strain rate of 4000 s^−1^, the EM sample shows the largest flow stress and the DM sample exhibits a moderate flow stress. Similarly, the WM sample shows the smallest flow stress.

The plastic deformability of the Ti-6321 alloy with different microstructures can be evaluated by the uniform plastic deformation strain. Under the strain rate of 1000 s^−1^, the EM sample shows the smallest uniform plastic deformation strain and the DM sample exhibits the largest uniform plastic deformation strain. The WM sample shows a moderate uniform plastic deformation strain. Under the strain rate of 2000 s^−1^, the EM sample shows a moderate uniform plastic deformation strain which is larger than that of the EM sample under the strain rate of 1000 s^−1^. The DM sample shows the largest uniform plastic deformation strain. The WM sample shows the smallest uniform plastic deformation strain. Under the strain rate of 3000 s^−1^, the EM sample shows a moderate uniform plastic deformation strain. The DM sample exhibits the largest uniform plastic deformation strain whereas the WM sample shows the smallest uniform plastic deformation strain. Under the strain rate of 4000 s^−1^, the EM sample shows the largest uniform plastic deformation strain. The DM sample shows a moderate uniform plastic deformation strain whereas the WM sample shows the smallest uniform plastic deformation strain.

The impact-absorbed energy, i.e., the energy absorbed during the process from plastic deformation initiation to the ultimate shear failure, can be used to evaluate the resistance of the materials to dynamic loading [22,23,24]. High impact-absorbed energy usually demonstrates a strong resistance to dynamic loading. The impact-absorbed energy (*E*) can be calculated as follows [25]:(1)$$E=\int_{\varepsilon_{1}}^{\varepsilon_{2}}\sigma d\varepsilon$$
where $\varepsilon_{1}$ represents the strain from which the plastic deformation initiated and $\varepsilon_{2}$ represents the strain where the shear failure occurred. σ is the mean dynamic flow stress which can be calculated from averaging the flow stress between $\varepsilon_{1}$ and $\varepsilon_{2}$. ε represents the maximum uniform plastic strain. The values of all parameters above are extracted from the true stress–strain curves of the Ti-6321 samples with different microstructures.

Figure 6 exhibits the impact-absorbed energy of Ti-6321 with EM, DM and WM under the strain rates of 1000 s^−1^, 2000 s^−1^, 3000 s^−1^ and 4000 s^−1^, which are calculated based on Equation (1). Under the strain rate of 1000 s^−1^, the DM sample shows the highest impact-absorbed energy and the WM sample exhibits higher impact-absorbed energy than the EM sample. When the strain rate is increased to 2000 s^−1^, the DM sample shows the highest impact-absorbed energy as well. However, the impact-absorbed energy of the EM sample is much higher than that of the WM sample. When the strain rate is increased to both 3000 s^−1^ and 4000 s^−1^, the DM sample shows the highest impact-absorbed energy whereas the WM sample exhibits the lowest impact-absorbed energy. In general, the Ti-6321 alloy with DM shows the highest impact-absorbed energy under each strain rate, indicating that the DM sample has the strongest resistance to dynamic loading as well as excellent dynamic mechanical properties. However, the Ti-6321 alloy with WM shows inferior resistance to dynamic loading which is demonstrated by the premature shear failure of the WM samples.

### 3.3. Adiabatic Shear Behaviors

The microstructure evolution of Ti-6321 samples was investigated after the dynamic pressure testing, in order to explore the adiabatic shear behaviors of the Ti-6321 alloy under dynamic loading. Figure 7 exhibits the microstructure of EM samples after dynamic pressure under the strain rates of 1000 s^−1^, 2000 s^−1^ and 3000 s^−1^. It can be seen that dynamic pressure under the strain rate of 1000 s^−1^ has scarcely influenced the microstructure of the EM sample, as shown in Figure 7a. Under the strain rate of 2000 s^−1^, α grains have also maintained the equiaxed shape, and initiation of the adiabatic shear band can be seen in the EM sample, as shown in Figure 7b. When the strain rate is increased to 3000 s^−1^, distinct adiabatic shear behaviors occur in the EM sample, as shown in Figure 7c. Moreover, microcracks are seen to be initiated and propagated along the adiabatic shear bands. Figure 8 shows the microstructure of DM samples after dynamic pressure under the strain rates of 1000 s^−1^, 2000 s^−1^ and 3000 s^−1^. It can be seen in Figure 8a,b that dynamic pressure under the strain rates of both 1000 s^−1^ and 2000 s^−1^ has hardly influenced the microstructure of the DM sample. However, adiabatic shear bands are formed in the EM sample when the strain rate is increased to 3000 s^−1^, as shown in Figure 8c. Figure 9 exhibits the microstructure of the WM samples after dynamic pressure under the strain rates of 1000 s^−1^, 2000 s^−1^ and 3000 s^−1^. Initiation of adiabatic shear bands can be seen in the WM sample under the strain rate of 1000 s^−1^, as the arrows show in Figure 9a. Several adiabatic shear bands can be seen in the WM sample with the strain rate increasing to 2000 s^−1^ and 3000 s^−1^, as shown in Figure 9b,c. The adiabatic shear bands penetrated into the prior β grains. Moreover, microcracks are seen to be initiated and propagated along the adiabatic shear bands.

### 3.4. Mechanisms for Adiabatic Shearing and Failure

To explore the mechanisms for adiabatic shearing in the Ti-6321 alloy, the DM sample with excellent dynamic mechanical properties was selected for further investigation. Figure 10 exhibits the general view and corresponding magnified regions of the adiabatic shear band in the DM sample deformed under the strain rate of 3000 s^−1^. The adiabatic shear band can be divided into three processes, i.e., initiation, expansion and fracture. It can be seen in the initiation section that the adiabatic shear band is relatively narrow, with a width of 4 μm~6 μm, as shown in Figure 10b. Subsequently, the adiabatic shear bands are seen to be propagated along the direction of the shearing stress. The width of adiabatic shear bands in the propagation section is 6 μm~12 μm which is wider than that in the initiation section, as shown in Figure 10b. Figure 10c exhibits the fracture section of the adiabatic shear bands with widths ranging from 12 μm to 17 μm. Moreover, microcracks are shown to be formed and propagated along the adiabatic shear band which may have induced the ultimate failure of the samples.

The SEM observation was performed according to the initiation, expansion and fracture regions of the adiabatic shear bands, for further investigation on the adiabatic shear band and the adjacent deformed microstructures of DM samples. Figure 11 shows the SEM photographs of the adiabatic shear band of the DM sample under the strain rate of 3000 s^−1^. In the initiation region, it can be seen that the primary α phase and transition β phase are compressed toward the shearing direction. The adiabatic shear bands have initially formed, containing the elongated boundaries of adiabatic shear bands which are similar to the adjacent microstructures, as shown in Figure 11a. Subsequently, the adiabatic shear bands are propagated along the shearing direction and the width of the adiabatic shear band is distinctly enlarged, as shown in Figure 11a. Moreover, the primary α phase, the lath-shaped α phase in transition to β phase, and the reserved β phase in the adiabatic shear bands is significantly compressed, as shown in Figure 11b. Clear boundaries between adiabatic shear bands can be distinguished from the adjacent microstructure. Figure 11c exhibits the fracture areas caused by the adiabatic shear bands. It can be seen that micro-voids have formed at the boundaries between adiabatic shear bands and the adjacent microstructures. Moreover, microcracks have formed and propagated along the boundaries of these shear bands. The formation of these micro-voids and microcracks can be attributed to the mismatch between the adiabatic shear bands and the adjacent microstructure [26]. As is well accepted, the formation of adiabatic shear bands is thought to dominate the dynamic fracture mode. The plastic deformation subjected to a high rate of deformation is converted into heat which cannot dissipate to the surroundings within the short time of the plastic flow process. The local increase in temperature causes thermal softening which, when greater than the work hardening, induces instability of the plastic deformation and adiabatic shear band nucleation sets in. The tendency for adiabatic shear band formation increases with increasing strain rates. The characteristics of the fracture show nucleation and coalescence of voids and cracks within the adiabatic shear bands, as seen in Figure 11c. Therefore, it is reasonable to believe the DM samples with low sensitivity to the adiabatic shearing can also show low sensitivity to microcrack nucleation.

## 4. Conclusions

The dynamic mechanical response and adiabatic shear behaviors of near-α Ti-6Al-3Nb-2Zr-1Mo (Ti-6321) alloy under impact loading were systematically investigated in the present study. The Ti-6321 samples with equiaxed microstructure (EM), duplex microstructure (DM) and Widmanstätten microstructure (WM) were prepared for dynamic mechanical properties testing under different strain rates. The main conclusions are as follows:(1)The Ti-6321 alloys with EM, DM and WM exhibited a strong strain-hardening effect. The flow stress of the Ti-6321 alloy was significantly improved with increased strain rate. However, the EM samples exhibited a larger flow stress than the DM and WM samples, demonstrating the stronger strain-hardening effect in EM samples.(2)The Ti-6321 alloys with EM, DM and WM exhibited different impact-absorbed energy under the same strain rate. The impact-absorbed energy of the DM sample was much higher than that of the EM and WM samples, indicating stronger resistance to dynamic pressure. The WM samples showed the lowest impact-absorbed energy when the strain rate was increased above 2000 s^−1^ which indicates an inferior resistance to dynamic pressure.(3)The Ti-6321 alloys with EM, DM and WM showed different sensitivity to adiabatic shearing. Adiabatic shearing occurred in the EM sample when the strain rate was increased to 2000 s^−1^. Adiabatic shear bands were formed in DM sample when the strain rate reached 3000 s^−1^. However, adiabatic shear bands were initiated in the WM sample under the strain rate of 1000 s^−1^. This demonstrates the DM sample’s low sensitivity to adiabatic shearing and the WM sample’s high sensitivity to adiabatic shearing.

## Figures and Tables

**Figure 1 materials-16-01406-f001:**
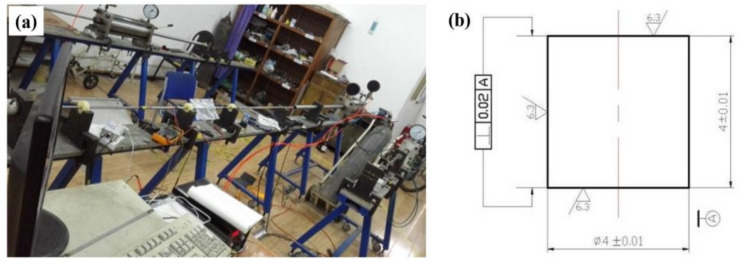
The split Hopkinson pressure equipment in (**a**) and the corresponding dimension of the sample used for dynamic performance tests in (**b**).

**Figure 2 materials-16-01406-f002:**
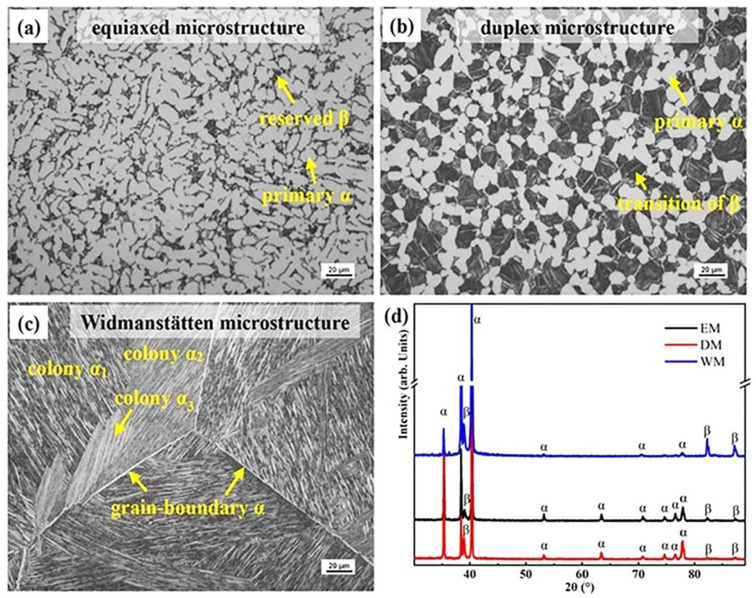
The microstructure of the Ti-6321 alloy by heat treatments at different temperatures: (**a**) 800 °C; (**b**) 960 °C; (**c**) 1020 °C; (**d**) the corresponding XRD patterns.

**Figure 3 materials-16-01406-f003:**
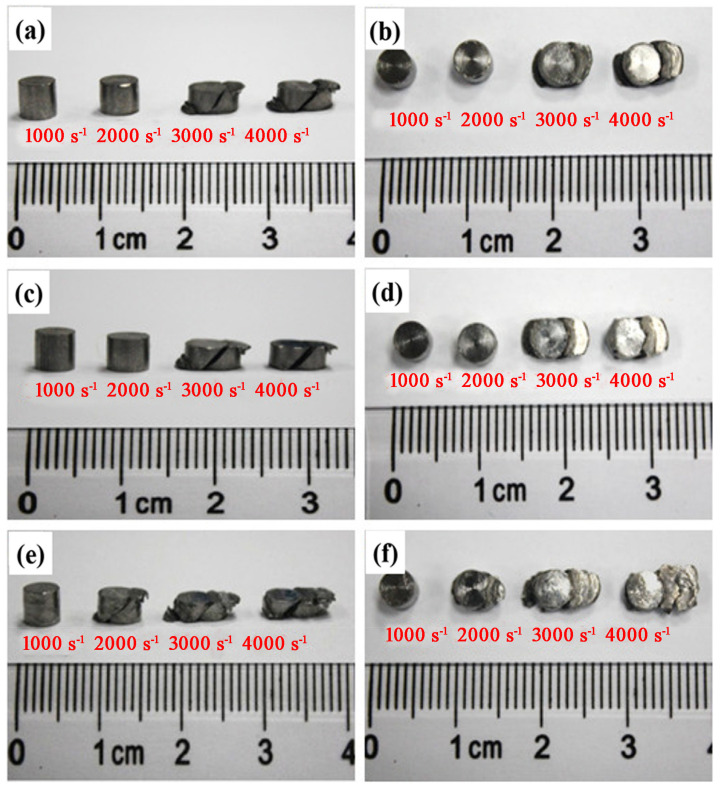
Ti-6321 samples with different microstructures at different strain rates: (**a**,**b**) EM; (**c**,**d**) DM; (**e**,**f**) WM.

**Figure 4 materials-16-01406-f004:**
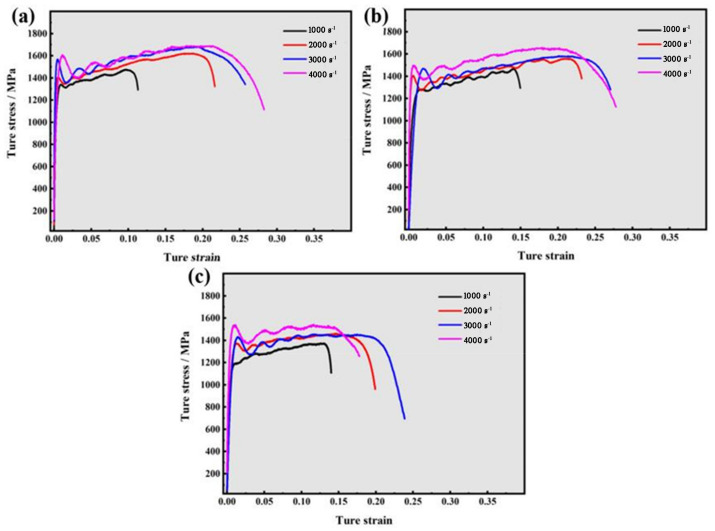
True stress–strain curves of Ti-6321 alloy with different microstructures: (**a**) EM; (**b**) DM; (**c**) WM.

**Figure 5 materials-16-01406-f005:**
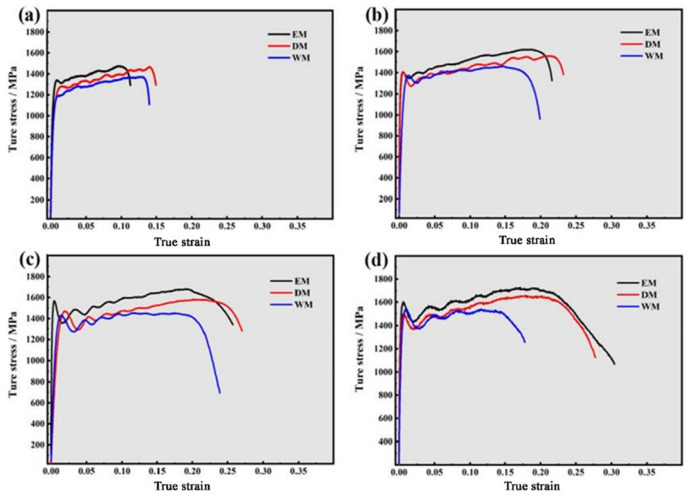
True stress–strain curves of samples with EM, DM and WM at different strain rates: (**a**) 1000 s^−1^; (**b**) 2000 s^−1^; (**c**) 3000 s^−1^; (**d**) 4000 s^−1^.

**Figure 6 materials-16-01406-f006:**
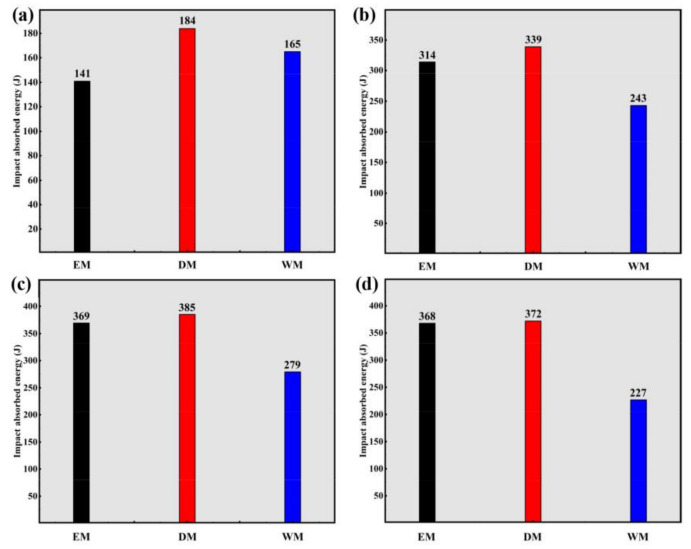
Impact-absorbed energy of Ti-6321 samples at different strain rates: (**a**) 1000 s^−1^; (**b**) 2000 s^−1^; (**c**) 3000 s^−1^; (**d**) 4000 s^−1^.

**Figure 7 materials-16-01406-f007:**
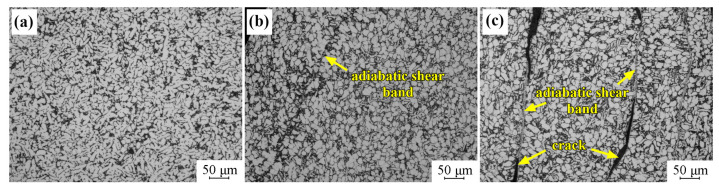
Microstructure of EM sample at different strain rates: (**a**) 1000 s^−1^; (**b**) 2000 s^−1^; (**c**) 3000 s^−1^.

**Figure 8 materials-16-01406-f008:**
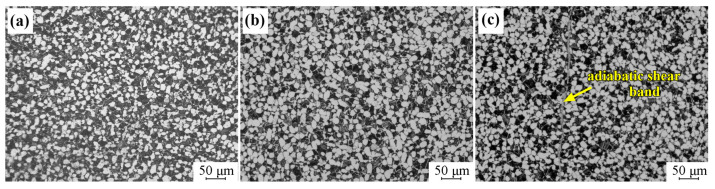
Microstructure of DM sample at different strain rates: (**a**) 1000 s^−1^; (**b**) 2000 s^−1^; (**c**) 3000 s^−1^.

**Figure 9 materials-16-01406-f009:**
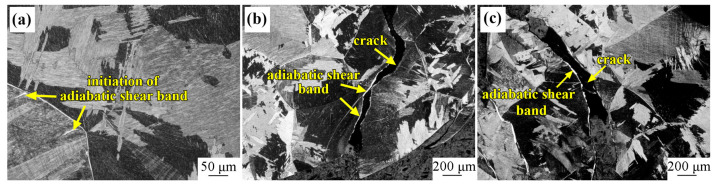
Microstructure of WM sample at different strain rates: (**a**) 1000 s^−1^; (**b**) 2000 s^−1^; (**c**) 3000 s^−1^.

**Figure 10 materials-16-01406-f010:**
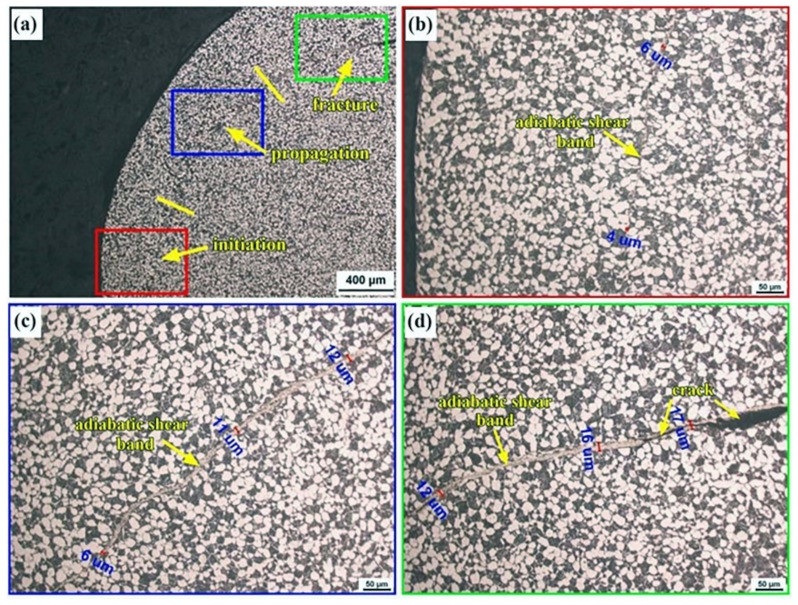
General view and magnified observations on the adiabatic shear band of the DM sample under the strain rate of 3000 s^−1^, (**a**) general view; (**b**) initiation; (**c**) expansion; (**d**) fracture.

**Figure 11 materials-16-01406-f011:**
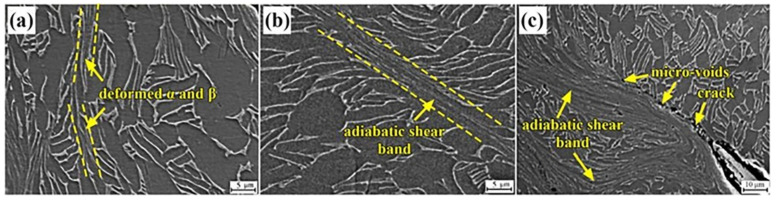
The SEM graphs of the adiabatic shear band in the DM sample under the strain rate of 3000 s^−1^: (**a**) initiation area; (**b**) expansion area; (**c**) fracture area.

**Table 1 materials-16-01406-t001:** The chemical composition of the Ti-6Al-3Nb-2Zr-1Mo (Ti-6321) alloy.

Element	Al	Nb	Zr	Mo	O	Ti
Content (wt.%)	6.13	3.02	1.95	0.97	0.09	balanced

**Table 2 materials-16-01406-t002:** Detailed heat treatments for Ti-6321 alloy.

No.	Heat Treatments
1	800 °C/1.5 h, air-cooling
2	960 °C/1.5 h, air-cooling
3	1020 °C/1.5 h, furnace-cooling

## Data Availability

Data available on request from the authors.

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
