# Peer review of "The Effects of Microstructure on the Dynamic Mechanical Response and Adiabatic Shearing Behaviors of a Near-α Ti-6Al-3Nb-2Zr-1Mo Alloy"

_materials, 2023, doi:10.3390/ma16041406_

Round 1

Reviewer 1 Report

Although the proposed manuscript is interesting, there are enough weaknesses that need to be improved. This based on the following:

·        The scope of the study is not well defined, the authors could better express it in the abstract

·        Line 16: In the abstract it is necessary to include the methodology and The use of acronyms in the abstract is confusing, you should omit them and in manuscript you can already do it in the different sections

·        Line 32: The introduction section needs to be further enriched, addressing the use of titanium alloys and their classification.

·        Line 79-80. The objective is not clear, it must be restructured again

·        Line 78-80: This paragraph should be deleted, it does not correspond to the objective or restructure the ideas; ……..”The relationships between microstructure and adiabatic shearing behaviors 78 in Ti-6321 alloy are discussed in details, in order to obtain a further understanding of the 79 relationship between the microstructure and mechanical properties”.

·        In section 2. experimental procedure, paragraph line 82-94 should be restructured for a better understanding.

·        Line 86: The chemical composition must be placed in the form of a table and must include its ranges for each element.

·        Table 1. The authors must indicate which is the reference for the selected parameters in the Heat treatments.

·        Line 104: Explain how the dynamic test strain rates were selected.

·        Line 108: The authors must indicate that to observe the samples under the microscope they made a metallographic preparation, not only indicate polishing and chemically atched.

·        In SEM which detector used backscattered electron (BSE) or secondary electron (SE).

·        Figure 2, 7, 8, 9 must say: OM microstructure.... and the magnification must be indicated (for example 500X).

·        In figure 3, while they are very distant and blurred, the millimeter scale is better observed.

·        Figure 4and 5, the mechanical behavior curves are plotted stress vs. strain, eliminate the word "True", it is necessary to correct the legends on the axes of the graphs.

·        check if the Impact absorbed energy can be better represented as the toughness of the material-

·        Figure 11 should be: SEM-BSE or SEM-Se morphology in the adiabatic ……….. and the magnification must be indicated.

·        The discussion of results should be enriched, only the authors describe the results but they are not discussed. The authors must relate the effect of the microstructure together with the role of the alloying elements of the material under study.

·        In the conclusions section, the first paragraph should be deleted. The authors must rewrite the conclusions and highlight the effect of the microstructure, alloy elements and the thermal treatment, since everything is related to a good mechanical behavior.

·         The authors present 22 references.  There is no self-plagiarism. the references section should be enriched

Author Response

Response to Reviewer #1:Although the proposed manuscript is interesting, there are enough weaknesses that need to be improved. This based on the following:

  1. The scope of the study is not well defined, the authors could better express it in the abstract.

Response: Thanks for your kind suggestions. The abstract has been modified and polished.

  1. Line 16: In the abstract it is necessary to include the methodology and The use of acronyms in the abstract is confusing, you should omit them and in manuscript you can already do it in the different sections.

Response: Thank you very much for your careful review and advice. We have added the description of methodology in the abstract. Besides, the abstract has been also modified according to this advice. All the modified parts have been marked in red in the revised manuscript.

  1. Line 32: The introduction section needs to be further enriched, addressing the use of titanium alloys and their classification.

Response: Thanks for your kind suggestions. The introduction has been enriched accordingly, with the addition of the use of titanium alloys and their classification. All the modified parts have been marked in red in the revised manuscript.

  1. Line 79-80. The objective is not clear, it must be restructured again.

Response: Thanks for your kind suggestions. We have revised the first conclusion in the revised manuscript according to your comments in the revised manuscript.

  1. Line 78-80: This paragraph should be deleted, it does not correspond to the objective or restructure the ideas;......”The relationships between microstructure and adiabatic shearing behaviors 78 in Ti-6321 alloy are discussed in details, in order to obtain a further understanding of the 79 relationship between the microstructure and mechanical properties”.

Response: Thank you very much for your careful review and advice. According to your comments, we have deleted 78-80 rows in the revised manuscript.

  1. In section 2. experimental procedure, paragraph line 82-94 should be restructured for a better understanding.

Response: Thanks for your kind suggestions. We have restructured paragraph line 82-94 according to your comments. All the modified parts have been marked in red in the revised manuscript.

  1. Line 86: The chemical composition must be placed in the form of a table and must include its ranges for each element.

Response: Thanks for your kind suggestions. According to your comments, we have listed the chemical composition in the form of table, as shown in Table 1 in the revised manuscript.

  1. Table 1. The authors must indicate which is the reference for the selected parameters in the heat treatments.

Response: Thank you very much for your careful review and advice. We have added the reference for the selected parameters in the heat treatments accordingly.

  1. Line 104: Explain how the dynamic test strain rates were selected.

Response: Thanks for your kind suggestions. We have added the references and the explanation of the dynamic tests accordingly, which have been marked in red in the revised manuscript.

  1. Line 108: The authors must indicate that to observe the samples under the microscope they made a metallographic preparation, not only indicate polishing and chemically atched.

Response: Thank you very much for your careful review and advice. We have added the detailed description of the corresponding metallographic preparation, which have been marked in red in the revised manuscript.

  1. In SEM which detector used backscattered electron (BSE) or secondary electron (SE).

Response: Thanks for your kind suggestions. We have revised the corresponding part, which have been marked in red in the revised manuscript.

  1. Figure 2, 7, 8, 9 must say: OM microstructure......and the magnification must be indicated (for example 500X).

Response: Thank you very much for your careful review and advice. We have added the magnification of OM microstructures.

  1. In figure 3, while they are very distant and blurred, the millimeter scale is better observed.

Response: Thanks for your kind suggestions. We have modified this section according to the advice, which have been marked in red in the revised manuscript.

  1. Figure 4 and 5, the mechanical behavior curves are plotted stress vs. strain, eliminate the word "True", it is necessary to correct the legends on the axes of the graphs.
  2. Response: Thank you very much for your careful review and advice. According to your comments in the revised manuscript, we have revised the legends on the axes of the graphs.
  3. check if the Impact absorbed energy can be better represented as the toughness of the material.

Response: Thanks for your kind suggestions.

  1. Figure 11 should be: SEM-BSE or SEM-Se morphology in the adiabatic......and the magnification must be indicated.

Response: Thank you very much for your careful review and advice. We have modified the figure caption of Figure 11. The magnification has also been added accordingly.

  1. The discussion of results should be enriched, only the authors describe the results but they are not discussed. The authors must relate the effect of the microstructure together with the role of the alloying elements of the material under study.

Response: Thanks for your kind suggestions. The discussion section has been modified and polished according to the advice, which have been marked in red in the revised manuscript.

  1. In the conclusions section, the first paragraph should be deleted. The authors must rewrite the conclusions and highlight the effect of the microstructure, alloy elements and the thermal treatment, since everything is related to a good mechanical behavior.

Response: Thank you very much for your careful review and advice. We have checked and modified the conclusion section, which have been marked in red in the revised manuscript.

  1. The authors present 22 references. There is no self-plagiarism. the references section should be enriched.

Response: Thanks for your kind suggestions. We have our references enriched accordingly, which have been marked in red in the revised manuscript.

Reviewer 2 Report

Figures 4 and 5. Vertical and horizontal axes' lables. Please, replace "ture" with word "true". 

Author Response

  1. The authors present 22 references. There is no self-plagiarism. the references section should be enriched.

Response: Thanks for your kind suggestions. We have our references enriched accordingly, which have been marked in red in the revised manuscript.

Reviewer 3 Report

The manuscript titled "Effects of microstructure on the dynamic mechanical response and adiabatic shearing behaviors of a near-α Ti-6Al-3Nb-2Zr-1Mo alloy" would be an interesting fit for publication in Materials but the article would need revision for the following comments before publication

1. In the introduction section of the manuscript, the authors should elaborate more on how the variations in composition, or metallurgical or mechanical changes brought about by phase changes of near-α Ti alloys can introduce strain hardening and its impact on adiabatic shearing. Such a discussion into the background of this study would make it more relevant.

2. In the XRD pictogram, the authors may find it worthwhile to calculate the volume fraction of the α and β phases at different temperatures. The authors mention that reserved β phase is found transformed into α phase during the air-cooling process once the temperature reaches over 1001 C, such a volumetric analysis would provide more quantitative evidence to the authors model presented.

3. The authors may also find it more relevant to include Thermocalc modelling to present the phase diagrams of the alloys at the different temperatures, it would be also worthwhile to calculate formation energy and thermodynamic properties for the different compositions.

4. Please explain section 3.3 the role of grain refinement on adiabatic shearing.

5. Please include a more comprehensive model for the formation of micro-cracks and micro-voids. Do the authors think that formation of intermetallic compounds as frequently seen in these near-α Ti alloys initiate the formation of micro-cracks during adiabatic shearing? Please explain in more detail.

Author Response

The manuscript titled "Effects of microstructure on the dynamic mechanical response and adiabatic shearing behaviors of a near-α Ti-6Al-3Nb-2Zr-1Mo alloy" would be an interesting fit for publication in Materials but the article would need revision for the following comments before publication.

  1. In the introduction section of the manuscript, the authors should elaborate more on how the variations in composition, or metallurgical or mechanical changes brought about by phase changes of near-α Ti alloys can introduce strain hardening and its impact on adiabatic shearing. Such a discussion into the background of this study would make it more relevant.

Response: Thanks for your kind suggestions. We have modified our introduction section according to the advice, which have been marked in red in the revised manuscript.

  1. In the XRD pictogram, the authors may find it worthwhile to calculate the volume fraction of the α and β phases at different temperatures. The authors mention that reserved β phase is found transformed into α phase during the air-cooling process once the temperature reaches over 1001 C, such a volumetric analysis would provide more quantitative evidence to the authors model presented.

Response: Thank you very much for your careful review and advice. We have modified this section according the advice, which have been marked in red in the revised manuscript.

  1. The authors may also find it more relevant to include Thermocalc modelling to present the phase diagrams of the alloys at the different temperatures, it would be also worthwhile to calculate formation energy and thermodynamic properties for the different compositions.

Response: Thanks for your kind suggestions.

  1. Please explain section 3.3 the role of grain refinement on adiabatic shearing.

Response: Thanks for your kind suggestions. We have modified the section 3.3 accordingly, which have been marked in red in the revised manuscript.

  1. Please include a more comprehensive model for the formation of micro-cracks and micro-voids. Do the authors think that formation of intermetallic compounds as frequently seen in these near-α Ti alloys initiate the formation of micro-cracks during adiabatic shearing? Please explain in more detail.

Response: Thank you very much for your careful review and advice. Detailed explanation has been added accordingly, which have been marked in red in the revised manuscript.

Reviewer 4 Report

This manuscript focuses dynamic mechanical response and adiabatic shear behaviors of near-α Ti-6Al-3Nb-2Zr- 16 1Mo (Ti-6321) alloy were systematically investigated by using the split Hopkinson pressure bar 17 (SHPB) tests. Consider, the following few points to attract the reader's attention and improve the article.

Author Response

This manuscript focuses dynamic mechanical response and adiabatic shear behaviors of near-α Ti-6Al-3Nb-2Zr- 16 1Mo (Ti-6321) alloy were systematically investigated by using the split Hopkinson pressure bar 17 (SHPB) tests. Consider, the following few points to attract the reader's attention and improve the article.1. There are few acronyms in the article. Even though the author has mentioned the acronyms in each paragraph, it’s better to note all the acronyms in a separate section after the conclusions section, for exampleNomenclatureDSC: Differential Scanning CalorimeterICP: Inductively Coupled Plasma Atomic Emission Spectrometry etc......

Response: Thank you very much for your careful review and advice. We have indicated all acronyms separately in the conclusion, according to your comments in the revised version.

1. The introduction section must be elaborated; Kindly add a Table below the introduction and summarize the critical work as follows Table X: A brief overview of the literature.

Response: Thank you very much for your careful review and advice. We have modified our introduction accordingly, which have been marked in red in the revised manuscript.

2. It’s better to add a flowchart in section 2 describing materials and methodology along with experimental detail for the readers better understanding.

Response: Thank you very much for your careful review and advice. We have added the flowchart in section 2 describing materials and methodology along with experimental detail.

3. Figure qualities must be improved, especially Figure No 2, 4, 5 and 6.

Response: Thank you very much for your careful review and advice. We have improved our Figure qualities.

4. Overall, the article is well written and kindly check for few Typos in few places.

Response: Thank you very much for your careful review and advice. According to your comments in the revised version, we have checked the whole article for typing errors and corrected them.

Round 2

Reviewer 1 Report

Figure 5 has serious errors. delete the word Ture and put True

Figure 11 should be: SEM-BSE or SEM-SE?

It is necessary to include a better discussion of results 

The authors have made the indicated corrections, after answering the last questions their work can be published.

Author Response

Response to Reviewer:

(1) Figure 5 has serious errors. delete the word Ture and put True

Response: Thanks for your kind suggestions. We have checked our manuscript and made the change in Figure 5.

(2) Figure 11 should be: SEM-BSE or SEM-SE?

Response: Thanks for your kind suggestions. The micrographs in Figure 11 are SEM-BSE, in order to distinguish the α phase and β phase.

(3) It is necessary to include a better discussion of results 

Response: Thanks for your kind suggestions. We have carefully modified section 3.4, as the discussion part. The modified parts are marked in red.

Reviewer 3 Report

The authors have updated the manuscript with the requested revisions and explanations, reviewer has no further questions

Author Response

Thanks for your kind suggestions. We have carefully modified the manuscript accordingly.

Reviewer 4 Report

The Article can be accepted 

Author Response

(The authors gave the same response as above.)
